# Optimal Neighborhood Selection for AR-ARCH Random Fields with Application to Mortality

**Paul Doukhan** [1], **Joseph Rynkiewicz** [2] **and Yahia Salhi** [3,*]

1 UMR 8088 Analyse, Géométrie et Modélisation, Université de Cergy-Pontoise, 2, Avenue Adolphe Chauvin, CEDEX, 95302 Cergy-Pontoise, France; doukhan@cyu.fr
2 Équipe SAMM, EA 4543, Université Paris I Panthéon-Sorbonne 90, Rue de Tolbiac, CEDEX 13, 75634 Paris, France; joseph.rynkiewicz@univ-paris1.fr
3 ISFA, LSAF EA2429, Univ Lyon, Université Lyon 1, 50 Avenue Tony Garnier, 69007 Lyon, France
* Correspondence: yahia.salhi@univ-lyon1.fr

**Abstract:** This article proposes an optimal and robust methodology for model selection. The model of interest is a parsimonious alternative framework for modeling the stochastic dynamics of mortality improvement rates introduced recently in the literature. The approach models mortality improvements using a random field specification with a given causal structure instead of the commonly used factor-based decomposition framework. It captures some well-documented stylized facts of mortality behavior including: dependencies among adjacent cohorts, the cohort effects, cross-generation correlations, and the conditional heteroskedasticity of mortality. Such a class of models is a generalization of the now widely used AR-ARCH models for univariate processes. A the framework is general, it was investigated and illustrated a simple variant called the three-level memory model. However, it is not clear which is the best parameterization to use for specific mortality uses. In this paper, we investigate the optimal model choice and parameter selection among potential and candidate models. More formally, we propose a methodology well-suited to such a random field able to select thebest model in the sense that the model is not only correct but also most economical among all thecorrectmodels. Formally, we show that a criterion based on a penalization of the log-likelihood, e.g., the using of the Bayesian Information Criterion, is consistent. Finally, we investigate the methodology based on Monte-Carlo experiments as well as real-world datasets.

**Keywords:** mortality rates; AR-ARCH random field; estimation; model selection; BIC

**MSC:** 60G70; 60G10; 60F99

## 1. Introduction

Recent advances in mortality modeling offer a wide variety of models to capture most of the stylized facts in human mortality dynamics. Most notably, there are the so-called factor-based models widely in use by practitioners, which have received increasing recognition from the actuarial community, e.g., [1] and the reference therein. Starting with the well-celebrated model in [2], most of the subsequent models rely on a factor-based decomposition of the mortality surface. These factors are intended to handle the complex patterns of mortality evolution over time and capture essentially the age, period, and cohort effects. Although these models are quite intuitive, their statistical properties are, however, not accurately understood. Moreover, the accuracy of such a framework has been widely discussed and shown that the latter fails, in many cases, to cope with some essential stylized facts inherent in mortality dynamics. Indeed, the cohort effect are of paramount importance when dealing with mortality in some countries, e.g., England and Wales, as well as the United States. For instance, the factor-based models attempted to incorporate a specific factor dependent on the year of birth to enhance the goodness-of-fit of models with only the age and period effect. Even if this undesired remaining diagonal effect is,

generally, accommodated, it is still unclear how such a cohort effect can be interpreted and identified, e.g., [3]. In contrast, these classic models are based on the assumption of the homoskedasticity of their residuals. However, the assumption of constant variance is always violated in practice as it is varying and does depend on the age and time.

A way to address these issues is to consider the surface of mortality improvements as a sole random field without any further assumption on the particular dependence structure, nor the factors driving its evolution. This is done in [4] using a parsimonious approach involving a generalization of the well-known AR-ARCH univariate processes; e.g., see [5]. The AR-ARCH for the mortality surface is a random field where the improvement rate dynamics are described as an autoregressive random field over a given adjacent neighborhood. The errors of these improvement rates are also described as an autoregressive conditional heteroskedastic model. Each part of the model is characterized by a neighborhood, which provides the number of lags in years and ages to be taken into account. The model is introduced in its general form, which is intended to accommodate to every population's own characteristics, i.e., cohort effect, cross-cohort dependence, as well as conditional heteroskedasticity. For more insight, in Section 3, we fully describe the model and provide some intuitions on its construction.

To estimate the parameters, ref. [4] used a quasi-maximum likelihood estimator (QMLE) and showed its consistency and asymptotic normality. For numerical illustration, they used a simple variant without any empirical basis. Therefore, in the current paper, we investigate the optimal model choice and neighborhood selection between the potential and candidate models. More formally, we propose a methodology well-suited to such a random field able to select the *best model* in the sense that the model is not only correct but also the most economical among all the *correct* models. Formally, in Section 3, after recalling some inference results in [4], we show that a criterion based on a penalization of the log-likelihood, e.g., the one using the Bayesian Information Criterion, is consistent.

The rest of the paper is organized as follows. In Section 2, after introducing the AR-ARCH random field model specification, we recall some theoretical results related to the existence of a stationary solution. In Section 3, we introduce the robust estimation procedure for the parameters based on the QMLE and recall some results on its consistency and asymptotic behavior. These will be crucial for defining a consistent procedure for model selection based a penalization of the QMLE. In this section, we will show that a specific penalization of the QMLE yields a consistent criterion for selecting the optimal neighborhoods. Finally, in Section 4, we assess the effectiveness of the procedure using Monte-Carlo experiments. Next, we use real-world data-sets to illustrate the procedure and compare the performance of the selected model to some benchmark models.

## 2. Random Fields Memory Models for Mortality Improvement Rates

### 2.1. Classic Mortality Models

A denoted by $m_{(a,t)}$, the crude death rate is at age $a$ and date $t$. Time is assumed to be measured in years so that calendar year $t$ has the meaning of the time interval $[t, t+1)$. For expository purpose and since we will be working with only a subset of historical data, we will henceforth re-index the observable ages by $a = 0, 1, \ldots, I-1$, and the observable dates by $t = 0, 1, \ldots, J-1$, where $I$ and $J$ are the number of ages and years, respectfully. The crude mortality rate $m_{(a,t)}$, among other actuarial quantities used to assess mortality, is of paramount importance when dealing with life insurance and pension liabilities. Its future evolution is generally modeled using the so-called factor-based models. Among others, there is the so-called Lee-Carter model [2]. In their seminal paper, Lee and Carter [2] postulated that the (log) mortality rates at different ages are captured by a common factor and an age-specific coefficient with respect to this common trend. More precisely, we have for any $a$ and $t$

$$\log m_{(a,t)} = \alpha_a + \beta_a \kappa_t + \epsilon_{(a,t)}, \text{ with } \epsilon_{(a,t)} \sim \mathcal{N}(0, \sigma) \tag{1}$$

where $\alpha_a$ is the time average level of $\log m_{(a,t)}$ at age $a$; $\kappa_t$ is the common factor also known as the period mortality effect; and $\beta_a$ is the age-specific sensitivity coefficient with respect

to $\kappa_t$. The development of the stochastic modeling of mortality was made on the basis of the above model. Indeed, the variety of models that have been introduced in the literature (see [6–9]), among others, were a generalization of the Lee-Carter model [2] in the sense that they generally include additional parameters, whether they are dependent on the age, the period, or even on the age of birth. These factors are intended to capture the complex patterns of mortality evolution over time, age, and cohort. As noted by [10], "the number and form of these types of effects is usually what distinguishes one model from another". However, some recent works show the limit of this mainstream approach, e.g., [3,10–12], among others. A major drawback of these models relates, in particular, to the assumption of homoskedastic error terms $\epsilon_{(a,t)}$. In fact, the assumption of constant variance is always violated: the observed logarithm of the central death rates is much more variable and the volatility is time varying; see, e.g., [13,14]. Many authors have already addressed the issue of using homoskedastic normal errors. For instance, [15] proposed using the Poisson assumption to model the number of deaths and [7] further allowed for overdispersion. Recently, Reference [16] developed an additional dispersion sub-model alongside the mean model for a joint modeling of the mean and dispersion. This line of literature is totally missing.

Furthermore, the mortality evolution is known to be related to the age of birth; see [17]. This is referred to as the cohort effect and translates the persistence of some shocks on mortality among cohorts. This is generally observed when plotting the residuals $\epsilon_{(a,t)}$ of models such as (1) as an apparent diagonal structure which requires additional univariate cohort processes in some countries. Such a phenomenon has lead to various extensions of the initial Lee–Carter model, e.g., [7,8] and the reference therein, by the inclusion of factors depending on the age of birth. The inclusion of additional univariate processes enhances the goodness-of-fit of the model but also overfits the model clearly and thus produces less reliable forecasts. Moreover, age–period–cohort models lack identifiability and suffer from the limited interpretability of its structure; see [3]. In addition, the heteroskedasticity of the mortality rates is not taken into account in such models as suggested by some recent empirical works; see [11,14,18], among others.

It is, of course, very important to tackle these limitations when considering a new modeling approach but it is also essential to take into account the dependence structure between adjacent cohorts. Indeed, some recent works and even common intuition point out the importance of cross-cohort correlation; see, e.g., [19,20]. In their empirical work, Loisel and Serant [19] showed that correlation among close generations is higher enough to be omitted. The same conclusions were drawn in the very recent work of [10]. It is worth mentioning that the Lee–Carter model in [2] in Equation (1) implicitly imposes a perfect dependence between mortality rates at the single age level. However, as noted by [19], common intuition suggests that correlation among close generations is high but should not necessarily be perfect. A first attempt to tackle this last stylized fact was done using a vector-based formulation of the mortality rates for a given range of ages. More specifically, it does consist of the modeling of the time-dependent vector $m_{(a,t)}$ for $a = 1, \ldots, I - 1$. For instance, refs. [19,21] were among the first to consider a model based on the study of the multivariate time series of the differences of *logits* of yearly mortality rates, taking into account inter-age correlations. Subsequent works focused on vector-autoregressive (VAR) models regarding a general framework without any assumption on the dependency between adjacent cohorts or with a given rigid dependency. For instance, in [10], the authors used a simple first-order vector autoregressive process, allowing for interaction between specific adjacent ages. This was applied to the residuals $\epsilon_{(a,t)}$ for a factor-based model to cope with the remaining non-stationary effect mentioned above. However, the dependency was limited to immediate generations. In other words, the residuals of population aged $a$ at time $t$ are only explained using the ones aged $a - 1$ and $a + 1$ during the same period. In the same line, [22] used the same idea but applied it directly to the (transformed) mortality rates. Similar approaches were also considered regarding joint modeling of multiple populations, e.g., [23,24].

### 2.2. Through a Random Field Framework

In contrast to this univariate factor-based framework, the authors in [4] approached the problem of modeling mortality rates by considering the whole surface as a sole random field without any further assumptions on the particular dependence structure, nor consideration of the factors driving its evolution. Thus, unlike the mainstream models, such a framework is intended to accommodate not only the cross-cohort dependence but also the conditional heteroskedasticity. This is amended by modeling the whole mortality (improvements) surface as a random field with a Markov property. The use of a random field for mortality modeling was, for example, considered in [25], wherein authors proposed a spatial random field model in which the covariance between adjacent cohorts was only a function of the distance between the ages but not of ages. However, the introduced framework is developed in continuous time and is inspired from the assumed similarities between the force of mortality and interest rates. It thus mainly aims at solving pricing problems for life annuities and longevity-linked securities, e.g., [26] for a discussion.

(i) Mortality Surface

In the sequel, we considered the process $X_s$ parameterized by the lattice points $s = (a,t)$ and defined as the *centered* mortality improvement rates $\mathrm{IR}_{(a,t)} = \log\left(m_{(a,t)}/m_{(a,t-1)}\right)$. Formally, we let

$$X_s = \mathrm{IR}_s - \overline{\mathrm{IR}}, \tag{2}$$

where $\overline{\mathrm{IR}}$ is the average improvement rate over $I \times J$. As noted before, empirical studies show that the differentiation of mortality rates (in the logarithmic scale) removes the non-stationarity; see, for example, [12]. This is also advocated by the models introduced above, as the time-dependent factors are described by random walks with a constant drift. However, these models assume that $\mathrm{IR}_s$ only depends on the observed age. Moreover, the conditional average of the improvement rates should not only depend on the age but also on the cohort, i.e., $t - a$, as well as on the rates experienced in adjacent generations. First, this is to allow for capturing the cohort effect, i.e., the persistent effect of mortality shocks in the same cohort. Second, including experience from neighbor generations allows for improving the assessment of the global mortality. Formally, this will account for the diffusion phenomenon well-known in demographic theory. Indeed, some changes in health risk behaviors are adopted first among some cohorts and then diffused through the population. We can refer to this as learning or diffusive effects. Therefore, in order to account for correlations in a natural way across generations, a Markov property for the random field $X_s$ is needed. Formally, we assume that $X_s$ has interaction with a finite number of neighbors arranged in any manner around the point $s = (a,t)$. This neighborhood is denoted by $V \subset \mathbb{N}^2 \setminus \{0\}$. Its shape is of paramount importance as it directly conditions the causality of the random field.

In Figure 1, at the lattice points $s = (a,t)$, here referred to as the point $(0,0)$, the neighborhood is depicted and corresponds to the grayed area, which obviously excludes the point $s$. This subset is causal in the sense of [5], that is, stable by addition.

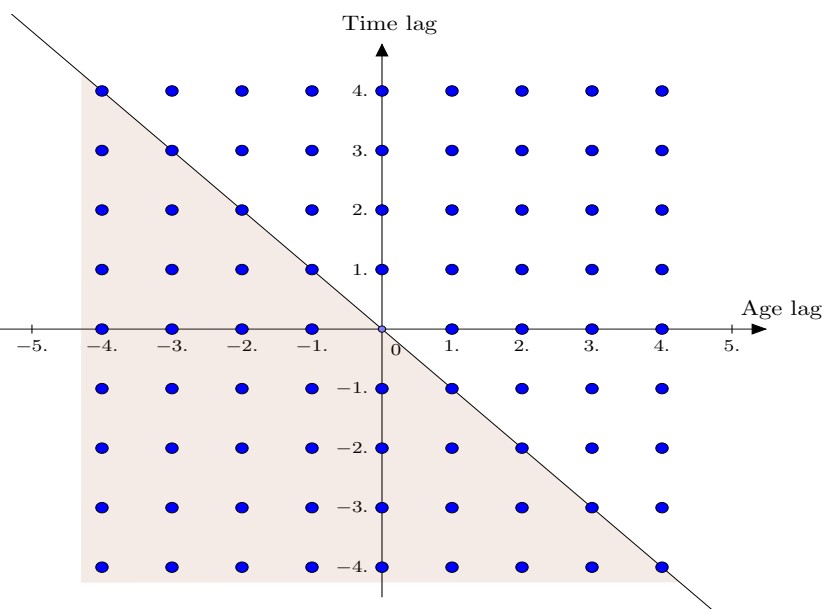

**Figure 1.** Causal random field: A bi-dimensional representation of the random field $X_s = \text{IR}_s - \overline{\text{IR}}$ with $s = (a, t)$. The grayed area represents the causal neighbor $V$ needed to characterize the evolution of $X_s$.

(ii) Random Field Memory Models

The starting point of the approach introduced in [4] is a formulation of the random field $X$ in the sense of [5]. More formally, let $V \subset \mathbb{N}^2 \setminus \{0\}$ be a given neighborhood; let $\Theta \subset \mathbb{R}^d$ be a set of possible parameters; let $d$ be a positive integer; consider $\xi = (\xi_s)_{s \in \mathbb{N}^2}$ as an independent and identically distributed (i.i.d.) random field; and let $F$ be a given parametric function taking values in $\mathbb{R}$: $F : \mathbb{R}^V \times \Theta \times \mathbb{R} \to \mathbb{R}$ (endowed with a suitable norm $\| \cdot \|$) and consider the solutions of the following equation:

$$X_s = F\Big((X_{s-v})_{v \in V}, \theta, \xi_s\Big), \quad \text{with } s \in \mathbb{N}^2. \tag{3}$$

For $Z \in \mathbb{R}^k$ ($k \geq 1$), denote $\|Z\|_p = (\mathbb{E}\|Z\|^p)^{1/p}$ for an integer $p \geq 1$. In the above equation, the set $V$, as discussed above, refers to a neighborhood of $X_s$ used to characterize its evolution. Recall that the set $V$ does not contain the point $(0,0)$, that is, $(s - v) \neq s$ for any $v \in V$. This is an abstract formulation in which $X_s$ is expressed in terms of its own past values and both the present and past values of a sequence of i.i.d. random variables. In the case where the set $V$ is causal, see, for example [5], the existence and uniqueness of a stationary solution of Equation (3) in Ł$^p$ ($p \geq 1$) rely on the contraction principle, which can be summarized in the following two properties:

A-1 $\|F(x_0, \theta, \xi)\|_p < \infty$ for some $x_0 \in \mathbb{R}^V$,

A-2 $\|F(x', \theta, \xi) - F(x, \theta, \xi)\|_p < \sum_{v \in V} \alpha_v \|x'_v - x_v\|$ for all $x = (x_v)_{v \in V}, x' = (x'_v)_{v \in V} \in \mathbb{R}^V$, where the coefficients $\alpha_v$ are such that $\sum_{v \in V} \alpha_v < 1$.

The model in Equation (3) is a general formulation of random field models with infinite interactions, which can be extended to the case where $X$ takes values in $\mathbb{R}^k$ and $F$ takes values in $\mathbb{R}^q$ for integers $k, q > 1$. These models are not necessarily Markov, linear, nor homoskedastic. Moreover, the inputs do not need additional distributional assumptions. It is thus an extension of ARMA random fields, which are special linear random fields; see, e.g., [27,28]. Such an extension yields a novel random field which is capable of taking into account the non-linearity and spatial dependency well-adapted in our context. In other words, we can fit a particular spatial model of the form (3) to such data to give an appealing tool for investigating both spatiality and non-Gaussianity patterns in the evolution of the mortality surface. Based on this abstract formulation, Doukhan el al. [4] proposed a specific

form for the function $F$ that is intended to tackle the various limitations encountered when using the factor-based approaches.

*2.3. AR-ARCH-Type Random Fields*

We let $s = (a, t) \in \mathbb{N}^2$ and consider subsets $V_1, V_2 \subset \mathbb{N}^2 \setminus \{0\}$. In view of the different stylized facts discussed above and regarding the aim of improving the mainstream approaches, ref. [4] considered a Markov property characterizing the conditional behavior of $X_s$. First, they proposed to decompose the specific features of the random field $X_s$ into a conditional mean $m_s$ and conditional variance $\sigma_s^2$ defined as

$$m_s = \mathbb{E}\Big[X_s|\,\{X_{(u,v)}; u < a, v \le t\}\Big], \text{ and } \sigma_s^2 = \mathbb{V}\mathrm{ar}\Big(X_s|\,\{X_{(u,v)}; u < a, v \le t\}\Big). \tag{4}$$

These conditional moments are then assumed to evolve as a patial autoregressive process, which has a structure very similar to the structure of the univariate AR model. More specifically, the conditional mean and variance, i.e., $m_s$ and $\sigma_s^2$ in Equation (4), are of the following form

$$\begin{cases} m_s &= \sum\limits_{v \in V_1} \beta_v X_{s-v}, \\ \sigma_s^2 &= \alpha_0 + \sum\limits_{v \in V_2} \alpha_v X_{s-v}^2, \end{cases} \tag{5}$$

where $(\beta_v)_{v \in V_1}$, $\alpha_0$, and $(\alpha_v)_{v \in V_2}$ are real positive coefficients. Here, $V_1$ and $V_2$ are the neighborhoods needed to characterize the evolution of the conditional mean and variance of each $X_s$ in terms of its own past values and both the present and past values of the adjacent cohorts. By doing so, we incorporate the observed heteroskedasticity in real-world datasets as well as the persistence of mortality shocks over adjacent cohorts. The main idea is that $\sigma_s^2$, the conditional variance of $X_s$, has a spatial autoregressive structure and is positively correlated to its own recent past and recent values of some adjacent cohorts defined by the set $V_2$. Therefore, the error terms are not equal across ages but rather they could be larger for some ages or ranges of ages than for others. Given these first two conditional moments, [4] introduced a random field model of the form:

$$X_s = m_s + \sigma_s \cdot \xi_s, \tag{6}$$

where $(\xi_s)_{s \in I \times J}$ is an independent and identically distributed (centered) random field. Here, the dynamics of $m_s$ and $\sigma_s$ are given in Equation (5). More specifically, the latter can be written in a more tractable form as follows

$$X_s = \xi_s \cdot \sqrt{\alpha_0 + \sum\limits_{v \in V_2} \alpha_v X_{s-v}^2} + \sum\limits_{v \in V_1} \beta_v X_{s-v}. \tag{7}$$

This is a class of conditionally heteroskedastic random fields with an autoregressive (AR) component where the innovation is defined from a standardized random field $\xi_s$ given as an autoregressive conditional heteroskedasticity (ARCH); see Equation (5). The combined model is referred to as the AR-ARCH random field and is a generalization of the now widely used AR-ARCH models for random processes; see [29]. As already mentioned, such a specification is in accordance with some recent results obtained for mortality series that exhibit conditional heteroskedasticity and conditional non-normality features, which are significant empirical long-term mortality structures; see, e.g., [11,12,30].

Using the notation from [5], which was recalled in the previous subsection, the parameters of the model are then denoted as $\theta = \Big(\alpha_0, (\alpha_v)_{v \in V_2}, (\beta_v)_{v \in V_1}\Big)$, where $V_2$ is the neighborhood for the conditional variance and $V_1$ is the the neighborhood characterizing the conditional mean. Note that the model in Equation (7) satisfies the assumptions

**A-1** and **A-2** needed to ensure the existence and uniqueness of a stationary solution. Indeed, note that the function $F$ is given by

$$F(x, \theta, z) = z \sqrt{\alpha_0 + \sum_{v \in V_2} \alpha_v x_v^2} + \sum_{v \in V_1} \beta_v x_v,$$

for $x = (x_s)_{s \in \mathbb{N}^2}$, $\theta = (\alpha_0, (\alpha_v)_{v \in V_2}, (\beta_s)_{v \in V_1})$, and $z \in \mathbb{R}$.

Then, by letting $\mu$ be the law of the centered i.i.d. random field $(\xi_t)_{t \in \mathbb{N}^2}$, Equation (6) can be expressed in the following form

$$X_s = H\big((\xi_{s-t})_{t \in \mathbb{N}^2}\big), \qquad \text{with} \qquad H \in \text{Ł}^p\big(\mathbb{R}^{\mathbb{N}^2}, \mu\big),$$

so that the conditions for the existence and uniqueness of a stationary solution can be written as

$$\|\xi_0\|_p < \infty, \quad \kappa_p \quad \equiv \quad \|\xi_0\|_p \sum_{s \in V_1} |\alpha_s| + \sum_{s \in V_2} |v_s| < 1, \quad p \geq 1. \tag{8}$$

For $p = 2$, a weaker condition than the previous one is given by

$$\|\xi_0\|_2 < \infty, \; \mathbb{E}\xi_0 = 0, \quad \kappa_2^2 \quad \equiv \quad \|\xi_0\|_2^2 \big(\sum_{s \in V_1} |\alpha_s|\big)^2 + \big(\sum_{s \in V_2} |v_s|\big)^2 < 1, \quad p = 2. \tag{9}$$

## 3. Estimation, Asymptotic Inference, and Model Selection

In this section, we study the maximum likelihood estimator (MLE) for the parameter $\theta$ in the AR-ARCH random field model in Equation (7). More precisely, we consider an approximation of the MLE, called the Quasi-Maximum Likelihood Estimator (QMLE), and prove its consistency together with the asymptotic normality. To this end, we assume throughout the sequel that the errors $\xi_s$ are an i.i.d. zero-mean and centered Gaussian random field.

### 3.1. Quasi-Maximum Likelihood Estimator (QMLE)

For the sake of simplicity, we consider that the observation are real numbers (i.e., taking values in $\mathbb{R}^k$ with $k = 1$) and the results exposed here can be extended to more general systems, but this remains a topic for future research. For a real function $g$ and an integer $p$, let us write $\|g(X)\|_p := \big(\mathbb{E}|g(X)|^p\big)^{\frac{1}{p}}$, which is the $\text{Ł}^p$ norm.

Suppose that the random field is observed for $\mathcal{I} = \{s{-}v, v \in V_1 \cup V_2\}$ (the initial states) and $s \in \mathcal{O} \subset \mathbb{N}^2$. Let us write $T$, which is the number of observations in $\mathcal{O}$, and assume that the following equation can always be written using $s \in \mathcal{O}$ and $s - v \in \mathcal{O} \cup \mathcal{I}$. Conditionally to $\mathcal{I}$, the quasi-log-likelihood of the model can be written as:

$$L_T(x_s, s \in \mathcal{O}; \theta) = \frac{1}{T}\left(\sum_{s \in \mathcal{O}} -\frac{1}{2} \ln\left(\alpha_0 + \sum_{v \in V_2} \alpha_v x_{s-v}^2\right) - \frac{\big(x_s - \sum_{v \in V_1} \beta_v x_{s-v}\big)^2}{2\big(\alpha_0 + \sum_{v \in V_2} \alpha_v x_{s-v}^2\big)}\right). \tag{10}$$

We will consider the estimator based on maximizing the above function (QMLE) over the set $\Theta$, which will be denoted as $\widehat{\theta}_T$, i.e.,

$$\widehat{\theta}_T = \arg\max_{\theta \in \Theta} L_T(x_s, s \in \mathcal{O}; \theta). \tag{11}$$

We say that $\widehat{\theta}_T$ is the QMLE estimator. In order to study the consistency of this estimator, we assume the following properties:

**Hypothesis 1 (H1).** *Finite second order moment, i.e.,* $\mathbb{E}(X_s^2) < \infty$.

**Hypothesis 2 (H2).** *The model is identifiable in the sense that for each set of parameters* $\big((\beta_v)_{\in V_1}, (\alpha_v)_{\in V_2}\big)$, *and* $\big((\beta_v')_{\in V_1}, (\alpha_v')_{\in V_2}\big)$, *we have.*

$$\left\| \sum_{v \in V_1} \beta_v X_{s-v} - \sum_{v \in V_1} \beta'_v X_{s-v} \right\|_2 = 0 \iff (\beta_v)_{v \in V_1} = (\beta'_v)_{v \in V_1},$$

*and*

$$\left( \alpha_0 + \sum_{v \in V_2} \alpha_v X^2_{s-v} \right) \left( \alpha'_0 + \sum_{v \in V_2} \alpha'_v X^2_{s-v} \right)^{-1} \stackrel{a.s.}{=} 1 \iff (\alpha_v)_{v \in V_2} = (\alpha'_v)_{v \in V_2}.$$

**Hypothesis 3 (H3).** *The set of possible parameters $\Theta$ is compact and the true parameter $\theta^0$ of the Model (7) belongs to the interior of $\Theta$.*

**Hypothesis 4 (H4).** $\mathbb{E}\left( X^4_s \right) < \infty.$

These are classic assumptions required for the consistency and asymptotic normality of the QMLE used in [4]. The hypothesis H1 assumes that the variance of the random field $X$ is finite, which is in line with the object of interest, i.e., mortality improvements. Furthermore, H2 ensures the identifiability of the model, which is a critical condition for the consistency as we may see later in this section. This will impose that the quasi-likelihood $L_T(x_s, s \in \mathcal{O}; \theta)$ has a unique maximum at the true parameter value $\theta^0$ over the compact set $\Theta$ from assumption H3. Based on these assumptions, [4] derived a first result which deals with the asymptotic behavior of the quasi-likelihood recalled in the following theorem.

**Theorem 1.** *For the model in Equation (7), under the assumptions H1, H2, and H3*

$$L(\theta) = \lim_{T \to \infty} L_T(x_s, s \in \mathcal{O}; \theta)$$

*exists for all $\theta \in \Theta$ and is uniquely maximized at $\theta^0$.*

This is the first step towards showing that the QMLE is consistent for the parameters of the jointly parametrized conditional mean and conditional variance. To this end, notice that the set of continuous functions $\mathcal{G}$ defined as

$$\mathcal{G} = \left\{ g\left( (x_s, (x_{s-v})_{v \in V_1}, (x_{s-v})_{v \in V_2}; \theta \right) = -\frac{1}{2} \ln \left( \alpha_0 + \sum_{v \in V_2} \alpha_v x^2_{s-v} \right) - \frac{\left( x_s - \sum_{v \in V_1} \beta_v x_{s-v} \right)^2}{2 \left( \alpha_0 + \sum_{v \in V_2} \alpha_v x^2_{s-v} \right)}, \theta \in \Theta \right\},$$

is Glivenko–Cantelli for the $\mathcal{L}_1-$ norm since the set of possible parameters $\Theta$ is a compact set, i.e., assumption H3. Then, in applying Theorem 5.7 of [4,31], we demonstrate that the QMLE is consistent and asymptotically normally distributed, provided that the random field has a finite fourth moment, i.e., assumption H4.

**Theorem 2 ([4]).**

(i) *Consistency. If the assumptions H1, H2, and H3 hold, then the QMLE estimator $\widehat{\theta}_T$ in Equation (11) is consistent in the sense that for each $(\beta_v)_{v \in V_1}, (\beta'_v)_{v \in V_1}, (\alpha_v)_{v \in V_2}$, and $(\alpha'_v)_{v \in V_2}$, we have*

$$\widehat{\theta}_T \xrightarrow{\mathbb{P}} \theta^0.$$

(ii) *Asymptotic normality. Under assumptions H1, H2, H3, and H4,*

$$\sqrt{T}\left( \widehat{\theta}_T - \theta^0 \right) \xrightarrow{\mathcal{L}} \mathcal{N}\left( 0, A_0^{-1} B_0 A_0^{-1} \right),$$

*where*

$$
\begin{cases}
A_0 = -\mathbb{E}\left[\dfrac{\partial^2 g(X_s, (X_{s-v})_{v \in V_1}, (X_{s-v})_{v \in V_2}; \theta^0)}{\partial \theta^2}\right], \\[4mm]
B_0 = \mathbb{E}\left[\dfrac{\partial g(X_s, (X_{s-v})_{v \in V_1}, (X_{s-v})_{v \in V_2}; \theta^0)}{\partial \theta}\left(\dfrac{\partial g(X_s, (X_{s-v})_{v \in V_1}, (X_{s-v})_{v \in V_2}; \theta^0)}{\partial \theta}\right)^T\right].
\end{cases}
\tag{12}
$$

Let us give some interpretations of the above results. (i) The asymptotic normal distribution result in Theorem 2 allows us to approximate the distribution of the estimated parameters $\theta$. This can be used, for instance, to obtain confidence intervals to conduct hypothesis testing. (ii) Results recalled in Theorem 2 are also of paramount importance when it comes to mortality forecasting. As the amount of data at our disposal is, generally, limited, the parameter estimates using the QMLE in (11) most inevitably will be subject to some degree of uncertainty. As demonstrated by [6], the parameter uncertainty forms a significant element of the uncertainty in forecasts of future mortality. Thus, Theorem 2 allows us to quantify this uncertainty on parameters based on their asymptotic distribution.

These different results deal mainly with the estimation of the parameters. Doukhan et al. [4] also introduced an estimation method based on moments matching. However, the latter is less effective than the one based on the likelihood maximization. Moreover, there is no statistical inference results for the moment method, which makes it less interesting for our application. In contrast, the question of model choice remained open. Indeed, the authors introduced an illustrative model considering two distinct neighborhoods for the conditional mean and conditional variance. Formally, the two neighborhoods considered in [4] are given by $V_1 = \{(1,1)\}$ and $V_2 = \{(1,0); (0,1)\}$. To some extent, numerical applications show that this simple variant is able to outperform classic models in predicting both mortality rates and residual life expectancies. However, the particular form of the neighborhoods does not have any empirical basis as the spatial dependence the between adjacent cohorts may change from one country to another. It is quite obvious that enlarging the sets $V_1$ and $V_2$ will fit the data better, in particular, if one compares the QMLE attained by each model. Moreover, there is no quantitative tool developed for such a class of models to optimally select the *best* model in the sense that a full feature of a given (mortality) population is represented. In general, when various models are plausible, the best model should be fitted within each of the competing types and compared using an information criterion. The latter encompasses the quasi-likelihood attained by each model but penalizes models that are over-parameterized. The choice of the penalty is of critical importance to ensure the consistency of the selected model. In other words, the convergence of the selected neighborhoods $V_1$ and $V_2$ to the *true* ones. In our case, in the literature, there is no theoretical results on the appropriate criterion to be used to ensure this consistency. Therefore, in the next section, we will show that, provided some technical assumptions on the asymptotic behavior of the penalty term, the consistency of the model selection is fulfilled by some of the classic procedures.

### 3.2. Optimal Model Selection

The previous subsection recalls some results of [4], dealing with the inference about the Model (6). As the model is general, it is not clear which is the best parameterization to use as there are a number of potential or candidate models. Here, being best is in the sense that the model is not only correct but also most economical, meaning that it is simplest among all the correct models. In this section, we deal with the problem of model selection and propose a methodology well-suited to the random field specification in Equation (6). The results recalled in Theorem 2 allow us to prove the tightness of the likelihood ratio test statistic (LRTS) and to prove the consistency of an estimator of the true neighborhoods for the conditional mean and conditional variance using an information criterion such as the Bayesian Information Criterion (BIC). First, we define the LRTS and provide its asymptotic behavior.

(i)  Likelihood ratio test statistic (LRTS)

Suppose we wish to test the null hypothesis $H_0 : \theta \in \Theta_0$ versus the alternative $H_1 : \theta \in \Theta_1$, where $\theta_0$ is a segment of an affine linear subspace of a segment of a bigger affine linear space $\Theta_1$. Write $\widehat{\theta}_{T,0}$ and $\widehat{\theta}_{T,1}$, which are the quasi-maximum likelihood estimators if the parameter set is $\Theta_0$ and $\Theta_1$, respectively. Here, we assume that the true value of the parameter $\theta_0$ is an inner point of $\Theta_0$ (so of $\Theta_1$). According to the definition of the quasi-log-likelihood (10), the LRTS denoted by $\Delta_T$ is given as follows:

$$\Delta_T = -2T \left( \sup_{\theta \in \Theta_1} L_T(x_s, s \in \mathcal{O}; \theta) - \sup_{\theta \in \Theta_0} L_T(x_s, s \in \mathcal{O}; \theta) \right). \tag{13}$$

Let us recall that a family of random sequences $\{Y_n, n = 1, 2, \ldots\}$ will be said to be $o_P(1)$ if for every $\delta > 0$ and $\varepsilon > 0$ there exists a constant $N(\delta, \varepsilon)$ such that $\mathbb{P}(|Y_n| < \varepsilon) \geq 1 - \delta$ for all $n \geq N(\delta, \varepsilon)$. A direct application of Theorem 2 and classic results on QMLEs (see [32]) yield:

$$\Delta_T = \sqrt{T} \left( \widehat{\theta}_{T,1} - \widehat{\theta}_{T,0} \right)^T A_0 \sqrt{T} \left( \widehat{\theta}_{T,1} - \widehat{\theta}_{T,0} \right) + o_P(1) \overset{T \to \infty}{\longrightarrow} \sum_{i=1}^{\dim(\Theta_1) - \dim(\Theta_0)} \lambda_i \chi_i^2(1). \tag{14}$$

where $(\chi_i^2(1))_{1 \leq i \leq \dim(\Theta_1) - \dim(\Theta_0)}$ are independent $\chi^2$ variables with one degree of freedom and $(\lambda_i)_{1 \leq i \leq \dim(\Theta_1) - \dim(\Theta_0)}$ are strictly positive real numbers. Note that there is no analytical computation of the distribution of a weighted sum of the $\chi^2$ variables. However, this result shows that the LRTS is tight (bounded in probability) for our model and we can assess the true model due to a suitable information criterion.

(ii)  Penalized QMLE

Let us denote $\mathcal{V}_1$ and $\mathcal{V}_2$, which are the maximal possible neighborhoods for the conditional mean and conditional variance of Model (7). A sub-model of

$$X_s = \xi_s \sqrt{\alpha_0 + \sum_{v \in \mathcal{V}_2} \alpha_v X_{s-v}^2} + \sum_{v \in \mathcal{V}_1} \beta_v X_{s-v},$$

will be specified by setting parameters $\alpha_v$ and $\beta_v$ to 0. Let us write $V^0 = V_1^0 \cup V_2^0$, which is the smallest neighborhood such that the Model (7) realizes the true regression function. For a fixed neighborhood $V = V_1 \cup V_2$, let $|V|$ be the number of elements of the neighborhood associated to non-zero parameters $\alpha_v$ or $\beta_v$ and write $\Theta_V$, which is the set of these non-zero parameters in Model (7). In order to optimally select the adequate model, we need to introduce another quantity that aims at penalizing the less *economical* models. Indeed, penalization is a classic approach to select a model. In short, penalization chooses the model, minimizing the sum of the empirical risk, i.e., how well the model fits data and some measure of the complexity of the model (called penalty). In our case, let us define the penalized QMLE as follows:

$$U_T(V) = T \times \sup_{\theta \in \Theta_V} L_T(x_s, s \in \mathcal{V}; \theta) - a_T(|V|), \tag{15}$$

and write $\widehat{V}$, which is the neighborhood maximizing (15). Here, $a_T(|V|)$ is a penalty, which is a function of $|V|$, i.e., the model complexity. In the sequel, the selection approach consists of assuming the existence of a true model of minimal size and aims at finding it. More formally, this corresponds to the existence of the set $V^0 = V_1^0 \cup V_2^0$ such that the model in Equation (7) realizes the true regression function. Then, we aim to localize these using a selection criterion. To this end, the following assumptions are needed to ensure that the selected neighborhoods tend almost surely (with probability one) to the true model when the dimension of the data goes to the infinity:

I-1    $\mathcal{V}_1$, $\mathcal{V}_2$ (maximal possible neighborhoods) are a finite set and $V_1^0 \subset \mathcal{V}_1$, $V_2^0 \subset \mathcal{V}_2$.

I-2    $a_T(.)$ is increasing, $a_T(k_1) - a_T(k_2)$ goes to infinity as $T$ tends to go to infinity as soon as $k_1 > k_2$, and $a_T(k)/T$ tends to 0 as $T$ tends to go to infinity for any $k \in \mathbb{N}$.

**Theorem 3.** *Under the assumptions of Theorem 2-(i), I-1 and I-2, $\widehat{V}$ converges in probability to $V^0$.*

**Proof.** Let us write $\mathcal{V} = \mathcal{V}_1 \cup \mathcal{V}_2$. Applying (14) yields

$$
\begin{aligned}
\mathbb{P}(|\widehat{V}| > |V^0|) &\leq \sum_{k=|V^0|+1}^{|\mathcal{V}|} \mathbb{P}\Big(U_T(k) \geq U_T(k^0)\Big) \\
&= \sum_{k=|V^0|+1}^{|\mathcal{V}|} \sum_{V,\,|V|=k} \mathbb{P}\Big(T \times \Big(\sup_{\theta \in \Theta_V} L_T(x_s, s \in \mathcal{V}; \theta), \\
&\qquad - \sup_{\theta \in \Theta^0} L_T(x_s, s \in \mathcal{V}; \theta)\Big) \geq a_T(|V|) - a_T(|V^0|)\Big) \xrightarrow{T \to \infty} 0,
\end{aligned}
$$

since $T \times \Big(\sup_{\theta \in \Theta_V} L_T(x_s, s \in \mathcal{V}; \theta) - \sup_{\theta \in \Theta^0} L_T(x_s, s \in \mathcal{V}; \theta)\Big)$ is bounded in probability and $a_T(|V|) - a_T(|V^0|)$ goes to go infinity as $T$ tends to go to infinity (Assumption I-2). Now, we can write

$$
\mathbb{P}(V^0 \not\subset \widehat{V}) \leq \sum_{V, V^0 \not\subset V} \mathbb{P}\left(\sup_{\theta \in \Theta_V} L_T(x_s, s \in \mathcal{V}; \theta) - \sup_{\theta \in \Theta^0} L_T(x_s, s \in \mathcal{V}; \theta) \geq \frac{a_T(|V|) - a_T(|V^0|)}{T}\right),
$$

and if $V^0 \not\subset \widehat{V}$, $\sup_{\theta \in \Theta_V} L_T(x_s, s \in \mathcal{V}; \theta) - \sup_{\theta \in \Theta^0} L_T(x_s, s \in \mathcal{V}; \theta)$ converges in probability to $\sup_{\theta \in \Theta_V} L(\theta) - L(\theta^0) < 0$, thus $\mathbb{P}(V^0 \not\subset \widehat{V}) \xrightarrow{T \to \infty} 0$ because $(a_T(|V|) - a_T(|V^0|))/T \xrightarrow{T \to \infty} 0$. Finally, $\widehat{V}$ converges in probability to a neighborhood such that $|\widehat{V}| \leq |V^0|$ and $V^0 \subset \widehat{V}$, hence $\widehat{V} \xrightarrow{T \to \infty} V^0$ in probability, which is the desired the result. $\square$

Note that it is easy to see that assumption I-2 is fulfilled by the Bayesian Information Criterion (BIC). Therefore, according to the above theorem, the BIC is a consistent selection criterion for our model.

(iii) Bayesian Information Criterion

As noted before, model selection is the choice of an element $V \subset \mathcal{V}_1 \cup \mathcal{V}_2$ (neighborhood) using the data records $X_s$ with $s \in \mathcal{O}$. To do this, we use the Bayesian Information Criterion (BIC) as a decision function that penalizes the quasi-log-likelihood at rate $\log T$ by the model dimension, i.e., $a_T(|V|) = |V| \log T$. More formally, we choose the model (neighborhoods) as

$$
\widehat{V} = \arg\min_{V \subset \mathcal{V}_1 \cup \mathcal{V}_2} U_T(V).
$$

This choice represents a compromise between the goodness-of-fit and a simple, interpretable model. In addition, due to Theorem 3, such a criteria selects the true model as $\widehat{V} \to V^0$ in probability. In addition, the use of the BIC allows for comparative analyses to the existing models. Indeed, most of the recent work on mortality modeling relies on this criteria to rank the models; see, e.g., [8,9], among others. However, we have to adjust the quasi-likelihood in order to include the information at the edges of the space $\mathcal{I}$. More specifically, one should let the parameters $\alpha_v$ and $\beta_v$ equal to zero for all the data that are not contributing to the QMLE.

(iv) Search Algorithm

The selection of the optimal model needs to browse over the maximal sets $\mathcal{V}_1$ and $\mathcal{V}_2$, looking into all the possible combination of points $s = (a, t) \in \mathcal{V}_1 \cup \mathcal{V}_2$ separately for the conditional mean and conditional variance. For instance, if the neighborhoods $\mathcal{V}_1$ and $\mathcal{V}_2$ consist of $n$ and $m$ elements, respectively, then the implementation of the selection procedure for this random field model has a $2^{n+m}$ complexity. In other terms, when $n = m = 3$, e.g., $V_1 = V_2 = \{(1,0), (1,1), (0,1)\}$, we need to test 64 models. Even if this is computationally very simple, the time complexity can make the model choice rapidly infeasible. In fact, when considering larger sets, the number of models grows drastically. Indeed, when one considers the maximal sets $\mathcal{V}_1$ and $\mathcal{V}_2$, as depicted in Figure 2 for instance, we should test $2^{16} = 65{,}536$ models, making this particular step very time-consuming. In fact, the selection procedure involves the optimization of the quasi-likelihood in order to estimate the parameters of each candidate model. For instance, if one relies on some statistical software, such as R, the non-linear optimization routines in most *packages* will be sufficient to find the values of the parameters $\theta$ that maximize the quasi-likelihood. This can be, for instance, implemented based on some derivative-free algorithms, such as the Nelder–Mead, using heuristics to search for the maximizers under the constraint of stationarity (8) or (9). However, these methods have excessive run-times and the characterization of the optimal model can take some days or even months. Therefore, in order to enhance this step, we can supply the optimization algorithm with the closed form formulas of the gradient and Hessian of the likelihood function $g$; see Appendix A. In our case, the execution time for implementing the neighborhood selection procedure on the maximal neighborhoods in Figure 2 using R was expected to take too much time. Therefore, we have solved this by considering parallel calculations and by making use of external compiled languages to speed up the selection routine.

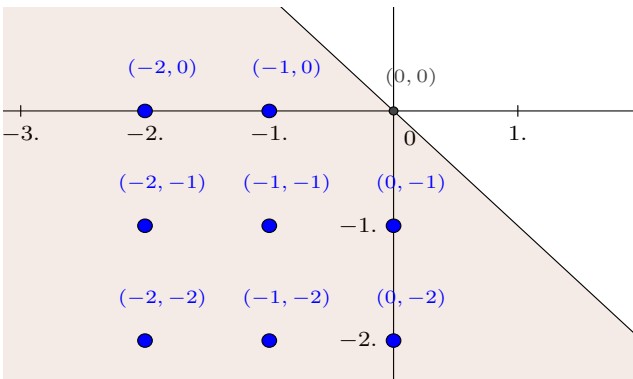

**Figure 2.** Maximal neighborhoods: Initial (maximal) candidate neighborhoods $\mathcal{V}_1 \equiv \mathcal{V}_2$ given as a collection of points $(i, j) \in \{(1,0); (1,1); (0,1); (1,2); (2,1); (2,2); (0,2); (2,0)\}$.

(v) Spatial Autocorrelation Function

In order to enhance the computation time, we can choose to reduce the dimension of the maximal neighborhoods. To this end, we will build an empirical procedure, which does not aim at robustness, on the basis of the well-celebrated Box and Jenkins procedure. The latter advocates for the inspection of the autocorrelation function both for the time-series as well as for their residuals. The pre-selection of the maximal lags relies on the confidence on the level of the autocorrelation. Therefore, in the sequel, we consider the generalization of the autocorrelation for random fields, called the spatial autocorrelation function. Formally, we denote the covariance function of the random field $X$ as $C(s, s - h) = \mathrm{Cov}(X_s, X_{s-h})$, where $s, s - h \in \mathcal{I}$. Since $X$ is stationary, we have $C(s, s + h) = C(0, h) = C_0(h)$ for all $s \in \mathcal{I}$ and $h$, and a stationary covariance function $C_0$. Now, let $S(h; \mathcal{I}) = \{s : s \in \mathcal{I}, s + h \in \mathcal{I}\}$ be the collection of pairs that are distanced by $h$ and denote

$|S(h; \mathcal{I})|$ by its cardinality, i.e., the number of distinct pairs in $S(h; \mathcal{I})$. Under the constant zero-mean assumption, the following statistic

$$\widehat{A}(h; \mathcal{I}) = \frac{1}{|S(h; \mathcal{I})|} \sum_{s \in S(h; \mathcal{I})} X_s X_{s-h}, \tag{16}$$

is an estimator of the covariance $C_0(h) = C(0, h)$ for a spatial lag $h \in \mathbb{R}_+^2$ and $\rho(h) = C_0(h)/C_0(0)$ is the estimator of the autocorrelation for the random field $X$. This functional will be used later on to empirically determine the candidate maximal sets $\mathcal{V}_1$ and $\mathcal{V}_2$. A more concise inspection of the asymptotic behavior of the autocorrelation in Equation (16) for an AR-ARCH random field is, however, beyond the scope of the current paper and will be explored in future work.

## 4. Numerical and Empirical Analyses

In this section, we investigate the performance of the BIC-based model selection. First, we run a Monte-Carlo experiment to assess the statistical properties developed in the previous section. Next, we consider a real-world dataset of mortality rates and explore the optimal neighborhood characterizing the mortality improvement rate dynamics. Hence, we forecast the future evolution of mortality and compare these outputs to classic models discussed in Section 2.1.

### 4.1. Simulation Study

In this section, we examine the finite sample properties of the model selection using Monte-Carlo simulations. With no loss of generality, we generate data from the model in Equation (6) with i.i.d. r.v. $\xi_s \sim \mathcal{N}(0, 1)$, given the specification of the neighborhoods in Figure 2 and an arbitrary set of parameters that fulfill the stationarity constraint. The Monte-Carlo analysis serves as an illustration of the robustness of the QMLE estimation as well as of the model selection. Using the notation of the previous sections, we investigate the impacts of varying the sample size $T$ on the performance of these procedures.

Building on the previous considerations, we set the sample size $T = I \times J = 3000$ ($I = 30$ ages and $J = 100$ years) and $30 \times 40$, and use 1000 replications. We then study the performance of the selection procedure. In Table 1, we report the outputs of this analysis. The two panels show how many times (in frequency) the BIC selected each model out of 1000 independent simulations for different sizes of the dataset. The true model is presented in *bold*, where the 1s correspond to the parameters belonging to the true neighborhood and 0s are those excluded. We reported the percentages of samples in which each competing model provides a better penalized fit than the other competing models. The model selection method performed reasonably well in identifying the true model, though its ability to recover the true model (in bold) increases with the sample size. For small-sized samples (lower part of Table 1), the model selection methods recovered at most 42.3%. When the sample size was large (upper part of Table 1), the model selection recovered at most 64.8%. We notice also that the number of competing models decreases with the sample size. Note that the consistency of the BIC selection criteria in Theorem 3 is achieved (the model selection method identifies (almost surely) the true neighborhoods) for big enough datasets, i.e., $I \times J \to \infty$, which is in line with the result shown in Table 1.

**Table 1.** Model selection: Monte-Carlo assessment of relative performance of the selection method across data sizes based on 1000 replications.

| | $V_2$ | | | | | $V_1$ | | | | |
|---|---|---|---|---|---|---|---|---|---|---|
| | (1,1) | (2,2) | (0,1) | (1,0) | $\alpha_0$ | (1,1) | (2,2) | (0,1) | (1,0) | |
| $I = 30, J = 100$ | 1 | 1 | 1 | 0 | 1 | 1 | 0 | 1 | 0 | 64.8% |
| | 1 | 1 | 0 | 1 | 1 | 1 | 0 | 1 | 0 | 11.1% |
| | 1 | 1 | 1 | 1 | 1 | 1 | 0 | 1 | 0 | 7.2% |
| | 1 | 1 | 1 | 0 | 1 | 1 | 0 | 1 | 1 | 6.1% |
| $I = 30, J = 40$ | 1 | 1 | 1 | 0 | 1 | 1 | 0 | 1 | 0 | 42.30% |
| | 1 | 1 | 1 | 1 | 1 | 1 | 0 | 1 | 1 | 12.00% |
| | 1 | 1 | 1 | 1 | 1 | 1 | 1 | 1 | 0 | 8.1% |
| | 1 | 1 | 1 | 1 | 1 | 1 | 0 | 1 | 0 | 8.1% |
| | 1 | 1 | 1 | 0 | 1 | 1 | 0 | 1 | 1 | 6.4% |
| | 1 | 1 | 0 | 0 | 1 | 1 | 0 | 1 | 0 | 5.4% |

In the light of the above remarks, particular attention should be paid to the implementation of this model for mortality dynamics. In fact, to be able to select the true model (neighborhoods), we need to have relatively large datasets to achieve the consistency requirement. However, we are then left with a challenging trade-off between enlarging the observed age and year periods or losing the consistency of the selection procedure. The former can undoubtedly be a good compromise to achieve the identification (almost surely) of the true neighborhoods but may alter the relevance of the fitted model. Indeed, including very old mortality improvement records would induce a misspecification of the future mortality decline. In fact, the mortality improvements were systematically lower at the beginning of the last century until the end of the second world war, expressed as the cohort effect. Therefore, in the following empirical analysis, we restrict the mortality data historical data and include the recent period. More precisely, we use the period of 1970–2016. The age band will be fixed to 55–89 to be in line with recent literature in mortality modeling, thus enabling readers to compare the estimation results to other papers more readily.

*4.2. Real-World Data Application*

In order to illustrate our model selection methodology, we consider the mortality data from the Human Mortality Database (HMD; the data was downloaded from www. mortality.org on 18 December 2016) for the male population of France (FRA), England and Wales (GBP), and the United States (USA). We have, at our disposal, the exposures and deaths sample, ranging from 1970 to 2016 and for 35 ages, namely 55 to 89 inclusive.

(i) Model Selection

Before proceeding to the selection of the optimal neighborhoods using the criterion in Section 3.2, we explore the autocorrelation function of the random field $X$ using the definition in Equation (16). The aim is to reduce the dimensionality of the maximal neighborhoods $\mathcal{V}_1$ and $\mathcal{V}_2$ by characterizing the two conditional moments. Using larger neighborhoods will not only require heavy machine calculation and thus take significant time to test all the candidate models but it is more likely to overfit the data. Therefore, we inspect the autocorrelation function to determine potential maximal lags in years and ages. This is motivated by the univariate time-series practice, which advocates for the examination of the so-called ACF and PACF plots These represent the autocorrelation function and the partial autocorrelation function in order to identify the maximum order (lag) of the models based on the characterization of the asymptotic behavior of the autocorrelation. In most univariate models, the autocorrelation decreases with a given rate, suggesting the nullity beyond a certain lag. The latter is used as the maximum lag. Unfortunately, in our case, there is no available asymptotic results for the behavior of the autocorrelation in (16), in particular, when the underlying random field is described using the AR-ARCH structure

in Equation (7). Developing a specific test for this particular model is challenging but goes beyond the scope of the current paper.

Here, we will only use a visual inspection of the autocorrelation function to determine the maximal neighborhoods $\mathcal{V}_1$ and $\mathcal{V}_2$ without any theoretical basis. If a test or the asymptotic behavior of the autocorrelation were available, one has to plot the confidence regions with a given probability and thus select the candidate models within these regions. Figure 3 shows the autocorrelation function (ACF) of the random field $X$ for the three populations (bottom panel) and the ACF of the squares of the random field (top panel). For each population, the squares are much more autocorrelated than the initial improvement, which is in line with the intuition behind the ARCH structure. The first orders' autocorrelation in year and ages lie in the interval $[0.4, 0.6]$ in the immediate neighborhoods and they gradually decline after some lags. These autocorrelations are sufficiently large to be taken into account locally. In view of the amount of data at our disposal, we may think that the computations of the autocorrelation for large lags are not consistent as few pairs $(X_s, X_{s-h})$ are available. To recap, these figures are in support of the initial hint on the autoregressive behavior of the mortality improvement rates together with the inter-generational dependence. Furthermore, the three populations show substantial evidence of the ARCH effects given the structure of the conditional variance, as judged by the autocorrelations of the squared process in the top panel of Figure 3. Indeed, the ARCH behavior is generally characterized by a little correlation in the process itself and by a profound correlation in the squared process; see Figure 3.

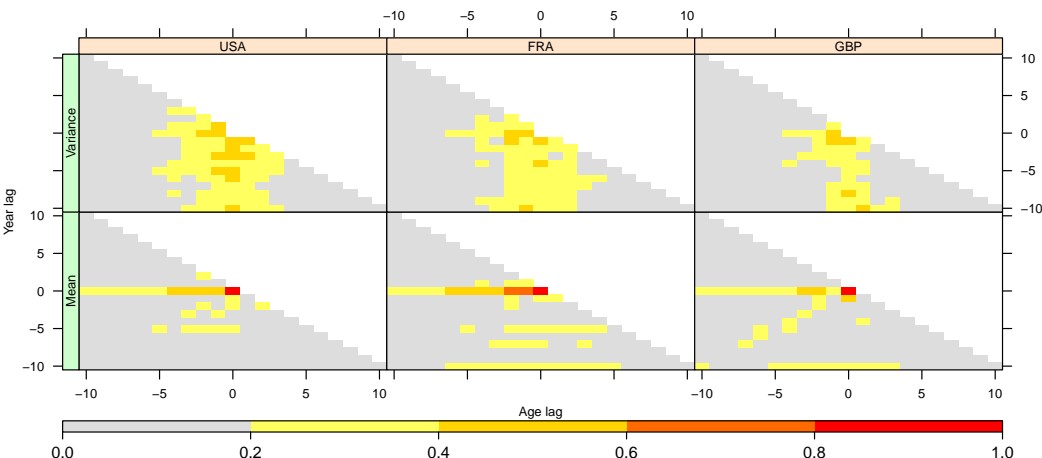

**Figure 3.** Autocorrelation: Autocorrelation of the random field and its squared values given in absolute value following the definition in Equation (16).

For the purposes of the present article and based on the considerations above, it is enough to consider a narrow characterization of each maximal neighborhood. Thus, for convenience and for the sake of applicability, the maximal neighborhoods used throughout the sequel are those displayed in Figure 2. Although this does not capture the whole dependence, as described in Figure 3, we will later check this section if the selected model conveniently captures the stylized behavior in each population's mortality.

In Figure 4, we report the results of the selection procedure. A set of $2^{16}$ models were investigated and the one maximizing the BIC criterion was selected. The parameters of the neighborhood with "—" do not belong to the optimal neighborhood. The remaining parameters are estimated and their values are reported.

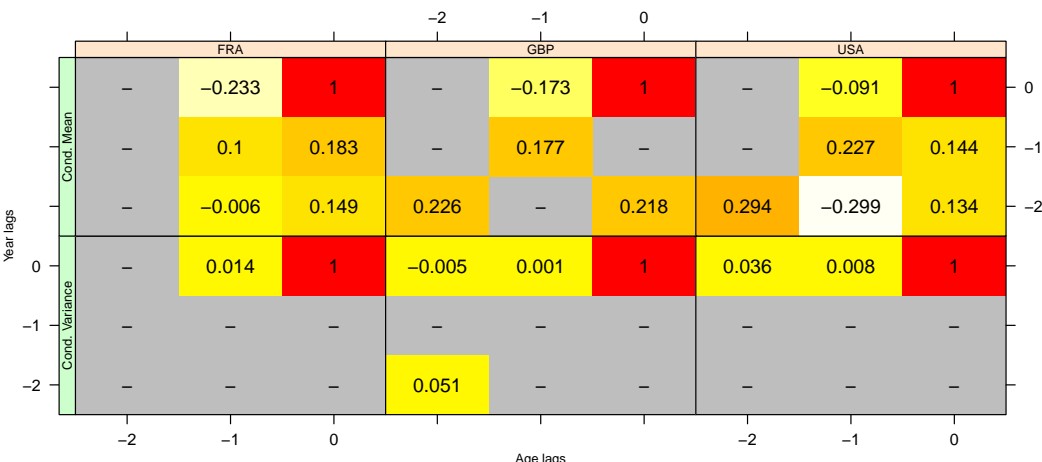

**Figure 4.** Model selection: Estimation of the parameters for the optimal model, maximizing the penalized QMLE. The grayed area corresponds to parameters that were not selected by the procedure and are excluded from the neighborhoods.

(ii) Diagnostic Checks

In order to assess the validity of the fitted models for these populations, we consider some standard diagnostics checks. Formally, we investigate the distributional assumption on $\zeta_s$ under the correct model. Therefore, given the estimation of the parameters $(\widehat{\alpha}_0, (\widehat{\alpha}_v)_{v \in V_2}, and (\widehat{\beta}_v)_{v \in V_1})$ for optimal neighborhoods $V_1$ and $V_2$ in Figure 4, we derive the corresponding implied residuals given by $\widehat{\zeta}_s = \widehat{m}_s / \widehat{\sigma}_s$, where the conditional mean and variance $\widehat{m}_s$ and $\widehat{\sigma}_s$ are given in Equation (5). These are assumed to behave as i.i.d. standard normal random fields. Figure A2 reports the QQ-plots of $\widehat{\zeta}_s$, depicting the empirical quantiles against those of a standard normal distribution. We notice that the plots are nearly straight lines except some outliers. However, the sample size is 2030, thus the outliers are a very small fraction of the data. Thus, a Gaussian distribution would be suitable for the residuals $\widehat{\zeta}_s$. Moreover, the histograms of the residuals, shown in Figure A3, appear to be normal with slight deviations at the tails for all populations. Further evidence may be obtained by applying normality tests such as those in [33]. These are some of the frequently used statistical tests for residual analysis. Failing these tests indicates that the residuals are not normal, which is an indication that the selected model is not consistent with the data. Results are displayed in Table 2, which shows the *p* values for some usual tests. For most of the tests, the residuals do validate the hypothesis of normality. Furthermore, we want to ensure that the residuals do not exhibit any serial correlation, which may violate the underlying assumption of independence. In Figure A1, we plot the spatial autocorrelation of the standardized residuals, as implied by the selected model using the definition in Equation (16). None of the plots provide any evidence of a persistent autocorrelation. Indeed, regardless of the lag in the age and year, the autocorrelation does not exceed 0.18 and is equal or close to 0 in most cases. That being said, we can conclude that the residuals for the three populations do satisfy the assumptions. Hence, the whole information embedded in each mortality surface is perfectly captured by the selected models and the remaining information consists only of white noise.

**Table 2.** Normality tests: We perform normality tests of the residuals $\widehat{\zeta}_s$ using the Shapiro–Wilk (SW) test, Anderson–Darling statistics (AD), the Cramér-von Mises test (CVM), Pearson-$\chi^2$ (PC) test, and the Shapiro–Francia (CF) test. The table shows the *p* values of the tests.

|        | SW     | AD     | CVM    | PC     | SF     |
|--------|--------|--------|--------|--------|--------|
| **USA** | 0.3428 | 0.7048 | 0.7991 | 0.8523 | 0.3947 |
| **FRA** | 0.6171 | 0.5212 | 0.5438 | 0.6279 | 0.4322 |
| **GBP** | 0.6114 | 0.7626 | 0.8548 | 0.9309 | 0.4827 |

However, we further examine the fitting of the selected models using residual *heatmaps*. The right panel of Figure 5 displays the residuals against the calendar year and age. In the left panel, we also depict the residuals of the Lee–Carter model fitted to the same data. These figures allow us to compare the goodness-of-fit behavior of the models. The plots of the Lee–Carter residuals indicate an obvious remaining structure not captured by the model. In particular, we observe clusters of high and low residuals, which is a specific feature that characterizes heteroscedastic fields. Indeed, this means that "large changes tend to be followed by large changes, of either sign, and small changes tend to be followed by small changes", which is a well-known effect in financial literature also called *volatility clustering*. In actuarial literature, such an effect is referred to as a cohort effect and represents the tendency of some group of cohorts to experience more rapid improvement mortality rates than generations born during either side of this period. However, this effect is captured by our model as we can see from Figure 5, which is also intended to inspect if any pattern still present was not perfectly captured by the selected models. Indeed, we observe that these residuals are reasonably *randomly distributed* along both the year and age dimensions. The heatmaps only show small traces of diagonal patterns around the cohorts born in 1920 and 1918 for the French population, but these are barely noticeable. These remaining *isolated* effects may be related to anomalies in the datasets. In fact, [34] highlighted the presence of such anomalies and errors in the data in period mortality tables in the form of isolated cohort effects for years of birth around the two world wars, that is, 1915, 1920, 1940, and 1945.

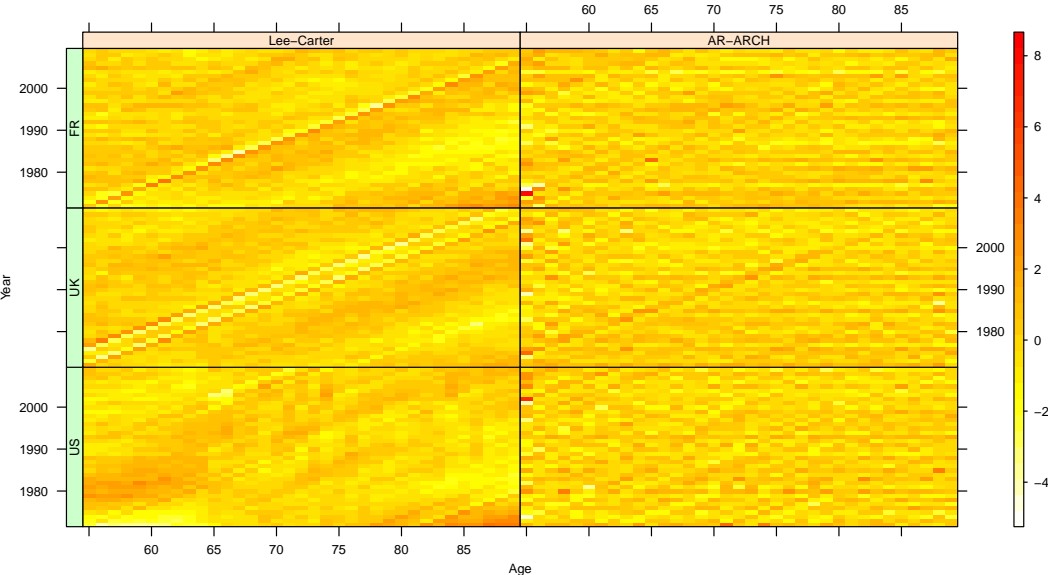

**Figure 5.** Standardized residuals: Heatmaps for the standardized residuals produced by the Lee–Carter (**left** panel) and the selected AR-ARCH model (**right** panel) for the French, England and Wales, and United States male populations.

We can thus conclude that all the selected models succeed to capture all the main features present in the initial mortality improvement rates.

(iii) Predictive Performance

The estimated models are used to predict the future evolution of mortality. We look at the mortality rates predicted by our model and the Lee–Carter model in Equation (1). Here, we only validate the performance based on an out-of-sample analysis instead of an in-sample inspection. This is because the benchmark model is over-parameterized and might undoubtedly outperform over the AR-ARCH.

In Figure 6, we display the 95% confidence intervals as well as the median prediction of the death rates from the two models for the selected ages 65, 75, and 85. First of all,

we observe that the projected mortality rates continue the historical downward trend for the considered populations. In comparison with the forecasts of the Lee–Carter model, we notice that the mortality rates predicted by our model show substantial differences at higher ages, i.e., 75 and 85, for instance. This phenomenon was also observed by [4] using a simple variant of the Model (6). For lower ages, at first sight, we see that the AR-ARCH model provides forecasts that are very close to the benchmark, which suggests that there is no tendency to a general out-performance at these age levels. This is due to the boundary condition, which makes the bias on the projection of the mortality rates made at lower ages spread over the diagonals. However, the impact of such a bias has to be minimized as the cohort effect and cross-generation dependency is limited at lower ages. This is why the model produces significantly different forecasts at high ages. Moreover, we observe that the evolution of mortality rates exhibits downward drifts comparable to the benchmark for all populations, with the exception of the UK population where our model anticipates a more drastic fall of mortality compared to the benchmark. Regarding the confidence intervals, we can observe that those related to the benchmark are quite narrow. This may be explained by the rigid structure of the Lee–Carter model as it was widely discussed in the literature. For instance, in [35], the Lee–Carter model was shown to generate overly narrow confidence intervals and generally result in an underestimation of the risk of more extreme outcomes. In contrast, the forecasting intervals produced by the AR-ARCH are substantially wider than those of the Lee–Carter model. The difference is even more pronounced for the higher ages, i.e., 75 and 85. This is mainly due, as discussed earlier, to the increasing uncertainty at higher ages, which is not captured by the benchmark that assumes a constant variance regardless of the age. The AR-ARCH allows us to cope with this drawback as the variance of the mortality improvements is also modeled as an autoregressive random field.

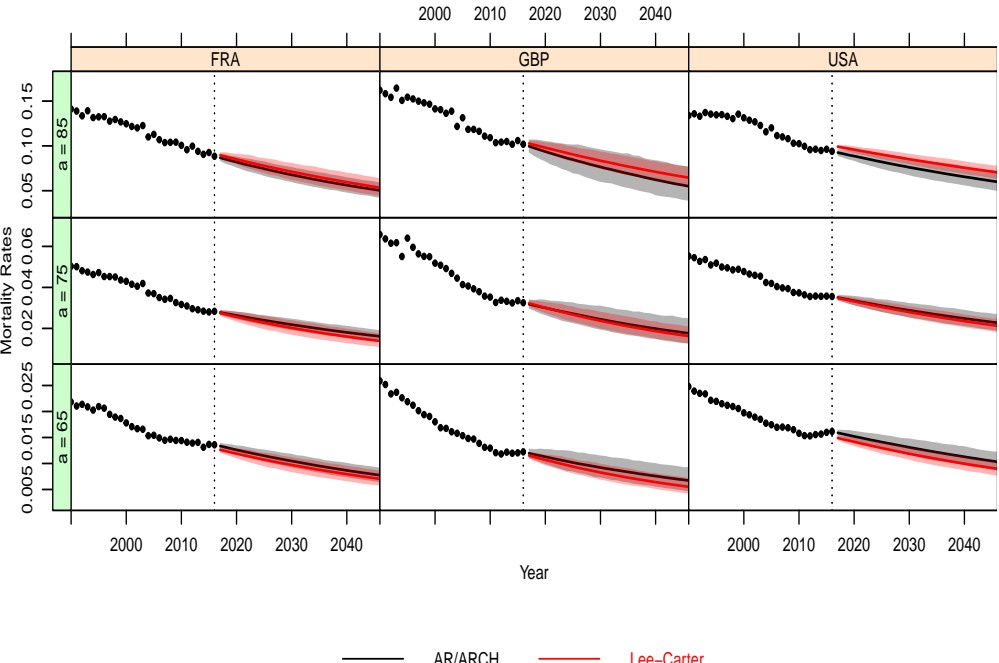

**Figure 6.** Mortality rates: Forecast mortality rates at age 65, 75, and 85 for France, England and Wales, and US populations produced by both the selected AR-ARCH (black) model and Lee–Carter (red) model. The 95% prediction intervals are also included.

For a closer examination, we use the forecasts provided by the out-of-sample analysis and derive the corresponding remaining period life expectancies. Figure 7 shows 95% confidence intervals for the life expectancies forecasts for ages 65, 75, and 85. We also depict with solid lines the median life expectancies of these projections. First, we can see that the

AR-ARCH random field model produces forecasts at least equivalent to the LC at lower ages. However, for higher ages, our model suggests a more increasing life expectancy compared to the Lee–Carter model. As soon as the confidence intervals are concerned, we remark that those related to the benchmark are quite narrow and underestimate the evolution of life expectancy. Therefore, the AR-ARCH forecasts should encompass a broader range of probable outcomes. The most striking feature of the AR-ARCH model is its ability to predict very high life expectancies at higher ages. It is then important to consider the biological reasonableness of such a behavior; see [6]. As noted by [36], this may be interpreted in terms of a limiting behavior of life expectancies. In other terms, the future life expectancy should not exceed a certain range. However, there is no common census of experts on such a limit. Therefore, in order to validate the behavior of the two models, we fit the models to data on the period ranging from 1970 to 1999 for each population and forecast the future improvement rates from year 2000 to 2016. Next, we recover the corresponding mortality rates as well as the life expectancies.

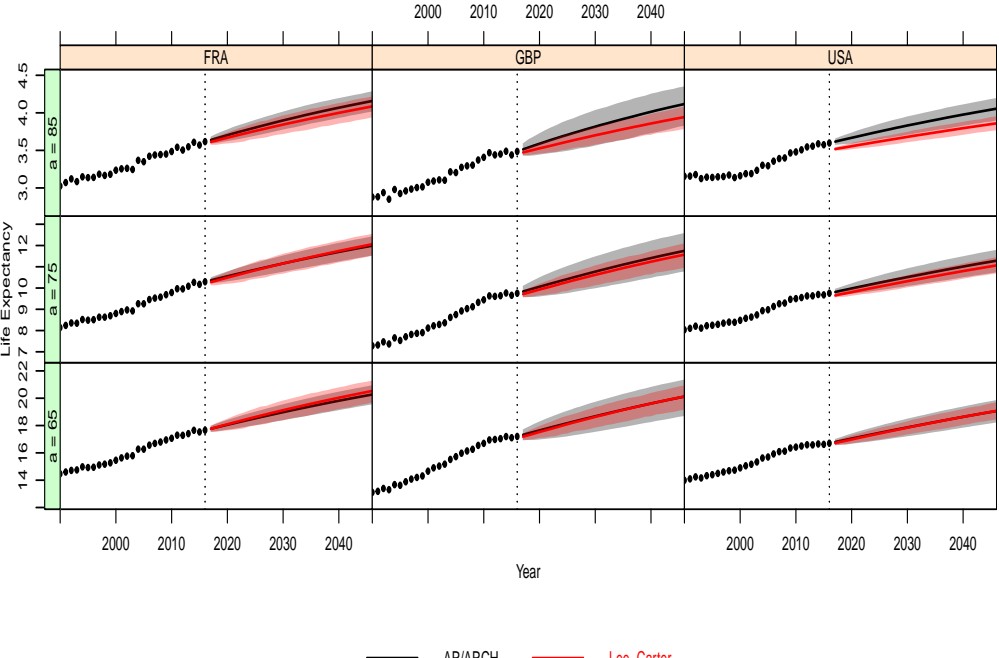

**Figure 7.** Life expectancy: Forecast life expectancy at age 65, 75, and 85 for the France, England and Wales, and US populations produced by both the selected AR-ARCH (black) model and Lee–Carter (red) model. The 95% prediction intervals are also included.

Besides the visual inspection of the forecast, we introduce some quantitative metrics to assess the forecast accuracy. In fact, as noted before, looking at Figure 7 may not reveal the actual performance over the whole mortality surface. Therefore, we consider an aggregate representative metric that summarizes the performance of each model. More precisely, we considered the mean absolute forecast error (MAFE) and root mean-squared forecast error (RMSFE). These criteria measured the closeness of the forecasts in comparison with the actual values of the variable being forecast regardless of the direction of the forecast errors. These are defined as follows:

$$\text{MAFE} = \frac{1}{|\mathcal{J}|} \sum_{s \in \mathcal{J}} \left| m_s - \widehat{m}_s \right|, \text{ and RMSFE} = \frac{1}{|\mathcal{J}|} \sum_{s \in \mathcal{J}} \sqrt{\left( m_s - \widehat{m}_s \right)^2},$$

where $\mathcal{J}$ is the set used to assess the forecasts and $|\mathcal{J}|$ is its size. Here, $m_s$ represents the actual holdout sample and $\widehat{m}_s$ the forecasts; see Figure A4 in Appendix B.2. In addition, we also consider a measurement for the confidence intervals' accuracy. As discussed

above, we wanted to check if the outputs of the considered model encompass sufficiently large amounts of observed quantities. Additionally, the model produced wide confidence intervals and one could question the coherence of such a behavior. In fact, this can be a drawback if the forecast interval is too wide compared to the realized scenarios. To do this, we introduce a scoring rule for evaluating the pointwise interval forecast accuracy defined by [37]. Therefore, with reference to the above analysis, we consider the symmetric 95% prediction interval and let $\widehat{m}_s^u$ and $\widehat{m}_s^l$ be its lower and upper bounds, respectively. Then, letting $s = (a, t)$, we define the scoring metric for a given forecast horizon $t$ and age $a$ as follows

$$\mathrm{IS}_{(a,t)}(95\%) = \left(\widehat{m}_s^u - \widehat{m}_s^l\right) + \frac{2}{5\%}\left(\widehat{m}_s^l - m_s\right)1_{m_s < \widehat{m}_s^l} + \frac{2}{5\%}(\widehat{m}_s - m_s^u)1_{m_s > \widehat{m}_s^u}.$$

As noted by [38], the optimal interval score is achieved when the actual realized mortality rate lies between the predicted bounds at the 95% level, with the distance between the upper and lower bounds being minimal. In order to summarize the above metric, as defined per age and forecast horizon, we consider the average score over the considered ages for each horizon. First, in Table 3, we compare the point forecast errors between the benchmarks and the optimal AR-ARCH model discussed so far, both for mortality rates $q_a$ and life expectancy forecasts. We can observe that the AR-ARCH model outperforms the other models in terms of point and forecast accuracy, as measured by the MAFE and RMSFE. This corroborates the above discussion and confirms that the model can behave in a similar fashion for some ages but outperforms, on average, both for life expectancy and probability rates. When comparing the interval forecast accuracy (see Figure 8), we observe that our model outperforms the others. In particular, the longer the forecast horizon, the larger the deviation between the AR-ARCH and benchmarks; see also Figures A4 and A5 in Appendix B.2. Moreover, we can remark that the score remains stable over the forecast horizon unlike the others whose score diverged over the horizon. However, for the French population, our model is very close to the Lee-Carter model [2]. This could be mainly due to the fact that the mortality dynamics for this particular population do not exhibit a clear cohort effect.

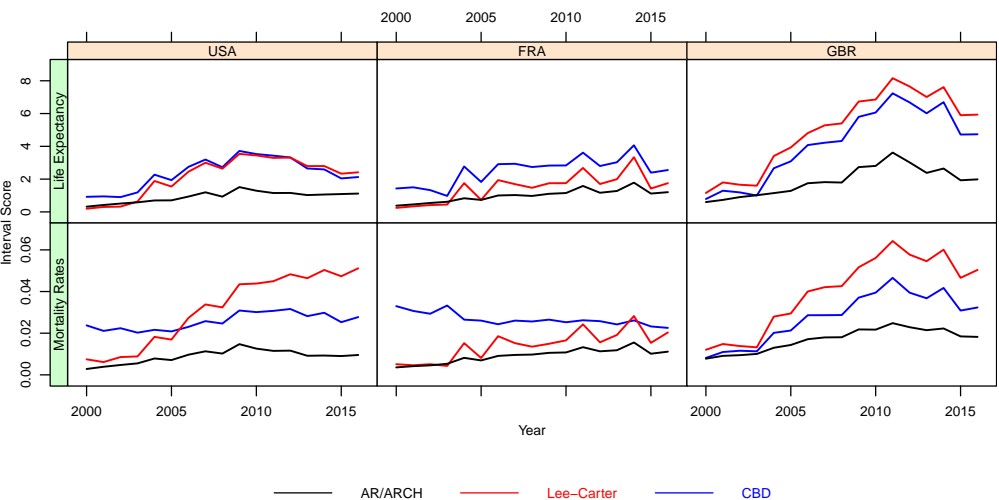

**Figure 8.** Confidence interval forecast score: The mean interval score across ages for different forecasting years for the LC, AR-ARCH, and CBD two-factor models.

**Table 3.** Point forecast accuracy: Forecast average errors between the LC, AR-ARCH, and CBD (see [6]) two-factor model both for mortality rates and life expectancies over the period of 2000–2016.

| | | | AR-ARCH | LC | CBD |
|---|---|---|---|---|---|
| **USA** | Mortality rate | **RMSFE** | $1.51 \times 10^{-5}$ | $5.30 \times 10^{-5}$ | $3.68 \times 10^{-5}$ |
| | | **MAFE** | $2.47 \times 10^{-3}$ | $3.99 \times 10^{-3}$ | $4.14 \times 10^{-3}$ |
| | Life expectancy | **RMSFE** | $1.13 \times 10^{-1}$ | $1.69 \times 10^{-1}$ | $2.72 \times 10^{-1}$ |
| | | **MAFE** | $2.75 \times 10^{-1}$ | $3.42 \times 10^{-1}$ | $4.34 \times 10^{-1}$ |
| **FRA** | Mortality rate | **RMSFE** | $1.07 \times 10^{-5}$ | $2.30 \times 10^{-5}$ | $5.04 \times 10^{-5}$ |
| | | **MAFE** | $2.19 \times 10^{-3}$ | $2.84 \times 10^{-3}$ | $4.49 \times 10^{-3}$ |
| | Life expectancy | **RMSFE** | $1.66 \times 10^{-1}$ | $2.11 \times 10^{-1}$ | $4.01 \times 10^{-1}$ |
| | | **MAFE** | $3.28 \times 10^{-1}$ | $3.79 \times 10^{-1}$ | $5.24 \times 10^{-1}$ |
| **GBR** | Mortality rate | **RMSFE** | $4.11 \times 10^{-5}$ | $8.34 \times 10^{-5}$ | $8.01 \times 10^{-5}$ |
| | | **MAFE** | $4.36 \times 10^{-3}$ | $5.83 \times 10^{-3}$ | $5.72 \times 10^{-3}$ |
| | Life expectancy | **RMSFE** | $7.29 \times 10^{-1}$ | $8.64 \times 10^{-1}$ | $8.70 \times 10^{-1}$ |
| | | **MAFE** | $7.03 \times 10^{-1}$ | $7.87 \times 10^{-1}$ | $7.89 \times 10^{-1}$ |

## 5. Concluding Remarks

In this paper, we addressed the problem of the optimal and robust selection of neighborhoods for the random field models introduced in [4]. The latter is a parsimonious alternative approach for modeling the stochastic dynamics of mortality improvements. We have shown that a procedure based on the penalization of the QMLE is asymptotically consistent in the sense that it selects the true model (neighborhoods) as far as the sample size goes to infinity. This particular behavior depends on the form of the penalty and we show that a classic BIC criterion fulfills the conditions that amend the consistency.

With this being shown, we numerically explored the procedure robustness and consistency for selecting the true model. For this, we drawed conclusions on the efficiency of the BIC-based penalization using Monte-Carlo experiments. We implemented the procedure in a simple case where the maximal neighborhoods are predefined. Next, we investigated the procedure using real-world datasets of mortality rates. As shown in [4] using even a simple model specification of the AR-ARCH structure, the selected model for the considered population corroborated the conclusions in [4]. In fact, our model outperforms the two classic factor-based approaches.

Finally, we should note that the AR-ARCH structure can be implemented to tackle the problems encountered when using factor-based approaches in order to characterize the evolution of the residuals. Indeed, based on the work of [10], which proposes a simple vector auto-regressive model (VAR) to model the residuals, one can hence consider the AR-ARCH as a parsimonious and coherent alternative for residuals' modeling. Indeed, as noted throughout the paper, the latter does not impose any predefined structure on the dependency of the underlying random field and does tackle the problem of the heteroskedasticity. These were not considered by the VAR model used in [10]. In contrast, other types of models may be considered in future works and [39] a reference for random processes. First, we will work out the case of the integer-valued models of the GLM type as examined in [40] since this naturally corresponds to count individuals of a certain age at a given date.

**Author Contributions:** All authors contributed equally to this work. All authors have read and agreed to the published version of the manuscript.

**Funding:** The work is supported by the Institut des Actuaires (French Institute of Actuaries) as well as the CY Initiative of Excellence (grant "Investissements d'Avenir" ANR-16-IDEX-0008), Project "EcoDep" PSI-AAP2020-0000000013. Y. Salhi's work is supported by the BNP Paribas Cardif Chair NINA "New Insurees, Next Actuaries". The views expressed herein are the author's owns and do not reflect those endorsed by BNP Paribas.

**Institutional Review Board Statement:** Not applicable.

**Informed Consent Statement:** Not applicable.

**Data Availability Statement:** The mortality data used in this article is freely available on Human Mortality Database (HMD), see www.mortality.org, accessed on 18 December 2016.

**Conflicts of Interest:** The authors declare no conflict of interest.

## Appendix A. The Gradient and the Hessian of the Quasi-Likelihood Function $g$

The derivatives of the function $g$ needed for the optimization problem can be derived explicitly after some algebra work and are given as follows. These appear also in Equation (12). First, the different elements of the gradient $\partial g / \partial \theta$ are given by:

$$\frac{\partial}{\partial \alpha_0} g(X_s, (X_{s-v})_{v \in V_1}, (X_{s-v})_{v \in V_2}; \theta^0) = -\frac{1}{2} \left( \frac{1}{\alpha_0^0 + \sum_{v \in V_1} \alpha_v^0 X_{s-v}^2} - \frac{\left( X_s - \sum_{v \in V_2} \beta_v^0 X_{s-v} \right)^2}{\left( \alpha_0^0 + \sum_{v \in V_1} \alpha_v^0 X_{s-v}^2 \right)^2} \right),$$

for $v$ in $V_1$

$$\frac{\partial}{\partial \alpha_v} g(X_s, (X_{s-v})_{v \in V_1}, (X_{s-v})_{v \in V_2}; \theta^0) = -\frac{1}{2} \left( \frac{X_{s-v}^2}{\alpha_0^0 + \sum_{v \in V_1} \alpha_v^0 X_{s-v}^2} - \frac{\left( X_s - \sum_{v \in V_2} \beta_v^0 X_{s-v} \right)^2 X_{s-v}^2}{\left( u_0^0 + \sum_{v \in V_1} \alpha_v^0 X_{s-v}^2 \right)^2} \right),$$

and for $v$ in $V_2$

$$\frac{\partial}{\partial \beta_v} g(X_s, (X_{s-v})_{v \in V_1}, (X_{s-v})_{v \in V_2}; \theta^0) = \frac{\left( X_s - \sum_{v \in V_2} \beta_v^0 X_{s-v} \right) X_{s-v}}{\alpha_0^0 + \sum_{v \in V_1} \alpha_v^0 X_{s-v}^2}.$$

Similarly, the elements of the Hessian matrix $\partial^2 g / \partial \theta^2$ are given by:

$$\frac{\partial^2}{\partial \alpha_0^2} g(X_s, (X_{s-v})_{v \in V_1}, (X_{s-v})_{v \in V_2}; \theta^0) = -\frac{1}{2} \left( -\frac{1}{\left( \alpha_0^0 + \sum_{v \in V_1} \alpha_v^0 X_{s-v}^2 \right)^2} + 2 \frac{\left( X_s - \sum_{v \in V_2} \beta_v^0 X_{s-v} \right)^2}{\left( \alpha_0^0 + \sum_{v \in V_1} \alpha_v^0 X_{s-v}^2 \right)^3} \right),$$

for $v$ in $V_1$

$$\frac{\partial^2}{\partial \alpha_0 \partial \alpha_v} g(X_s, (X_{s-v})_{v \in V_1}, (X_{s-v})_{v \in V_2}; \theta^0) = -\frac{1}{2} \left( -\frac{X_{s-v}^2}{\left( \alpha_0^0 + \sum_{v \in V_1} \alpha_v^0 X_{s-v}^2 \right)^2} + 2 \frac{\left( X_s - \sum_{v \in V_2} \beta_v^0 x_{s-v} \right)^2 X_{s-v}^2}{\left( \alpha_0^0 + \sum_{v \in V_1} \alpha_v^0 X_{s-v}^2 \right)^3} \right),$$

for $v$ in $V_2$

$$\frac{\partial^2}{\partial \alpha_0 \partial \beta_v} g(X_s, (X_{s-v})_{v \in V_1}, (X_{s-v})_{v \in V_2}; \theta^0) = \frac{-2 \left( X_s - \sum_{v \in V_2} \beta_v^0 X_{s-v} \right) X_{s-v}}{\left( \alpha_0^0 + \sum_{v \in V_1} \alpha_v^0 X_{s-v}^2 \right)^2},$$

for $v$ and $v'$ in $V_1$

$$\frac{\partial^2}{\partial \alpha_v \partial \alpha_{v'}} g(X_s, (X_{s-v})_{v \in V_1}, (X_{s-v})_{v \in V_2}; \theta^0) = -\frac{1}{2} \left( -\frac{X_{s-v}^2 X_{s-v'}^2}{\left( \alpha_0^0 + \sum_{v \in V_1} \alpha_v^0 X_{s-v}^2 \right)^2} + 2 \frac{\left( X_s - \sum_{v \in V_2} \beta_v^0 x_{s-v} \right)^2 X_{s-v}^2 X_{s-v'}^2}{\left( \alpha_0^0 + \sum_{v \in V_1} \alpha_v^0 X_{s-v}^2 \right)^3} \right),$$

for $v$ in $V_1$ and $v'$ in $V_2$

$$\frac{\partial^2}{\partial \alpha_v \partial \beta_{v'}} g(X_s, (X_{s-v})_{v \in V_1}, (X_{s-v})_{v \in V_2}; \theta^0) = -\frac{\left(X_s - \sum_{v \in V_2} \beta_v^0 X_{s-v}\right) X_{s-v}^2 X_{s-v'}}{\left(\alpha_0^0 + \sum_{v \in V_1} \alpha_v^0 X_{s-v}^2\right)^2},$$

and for $v$ and $v'$ in $V_2$

$$\frac{\partial^2}{\partial \beta_v \partial \beta_{v'}} g(X_s, (X_{s-v})_{v \in V_1}, (X_{s-v})_{v \in V_2}; \theta^0) = -\frac{X_{s-v} X_{s-v'}}{\alpha_0^0 + \sum_{v \in V_1} \alpha_v^0 X_{s-v}^2}.$$

## Appendix B. Additional Figures

*Appendix B.1. Diagnostic Checks of Models' Residuals*

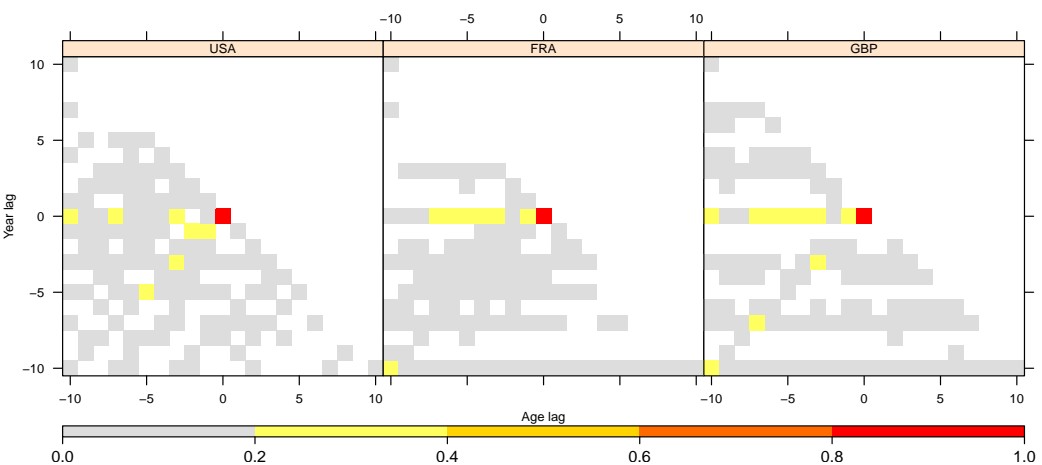

**Figure A1.** Autocorrelation: Inspection of spatial autocorrelation of the residuals $\widehat{\zeta}_s$ using the definition of the spatial autocorrelation function given in Equation (16).

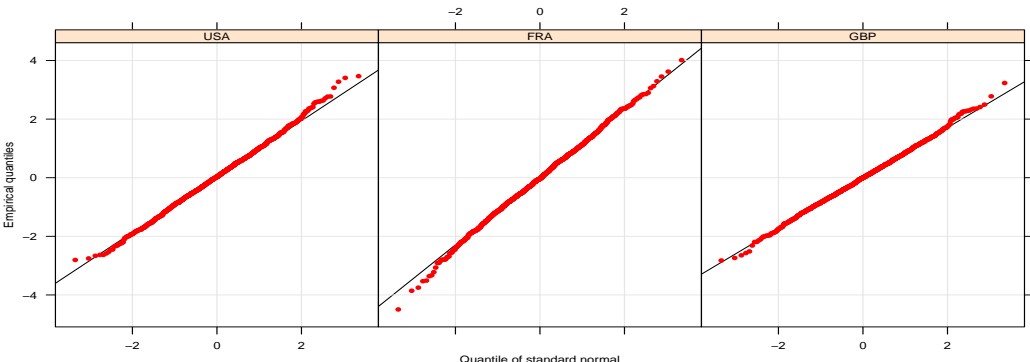

**Figure A2.** QQ-plots: Visual inspection of the normality of the residuals. Drawing of the QQ-plots is based on 100 Monte-Carlo simulations and comparisons between the obtained empirical quantiles of $\tilde{\zeta}_s$ with the theoretical quantiles of a centered Gaussian i.i.d. random field.

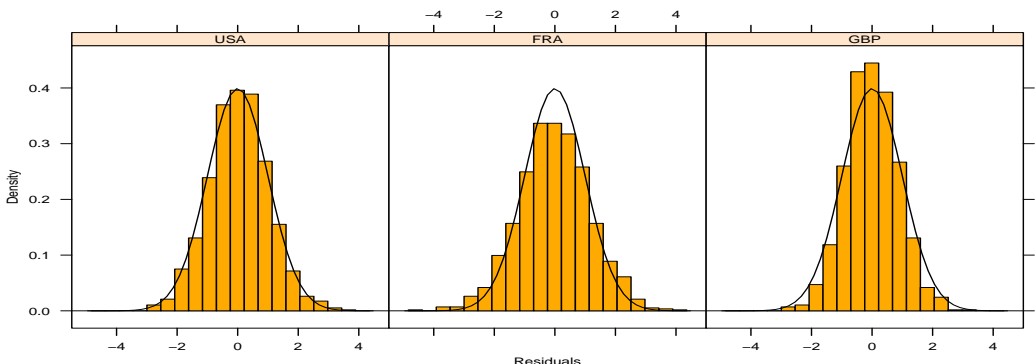

**Figure A3.** Histograms of residuals: Comparison of the historical residuals' densities (histograms) and a standard Gaussian density (lines).

*Appendix B.2. Out-of-Sample Analysis and Predictive Performance*

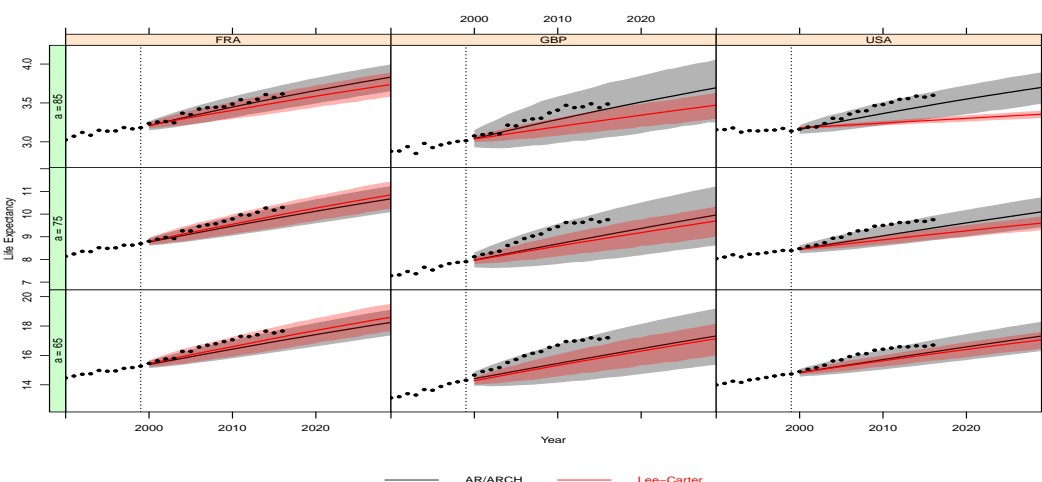

**Figure A4.** Life expectancy: Forecast life expectancy over the period of 2000–2030 at ages 65, 75, and 85 for the France, England and Wales, and US populations produced by both the selected AR-ARCH (black) model and Lee–Carter (red) model. The 95% prediction intervals are also included. The crude mortality over the observed period is also presented (black circles).

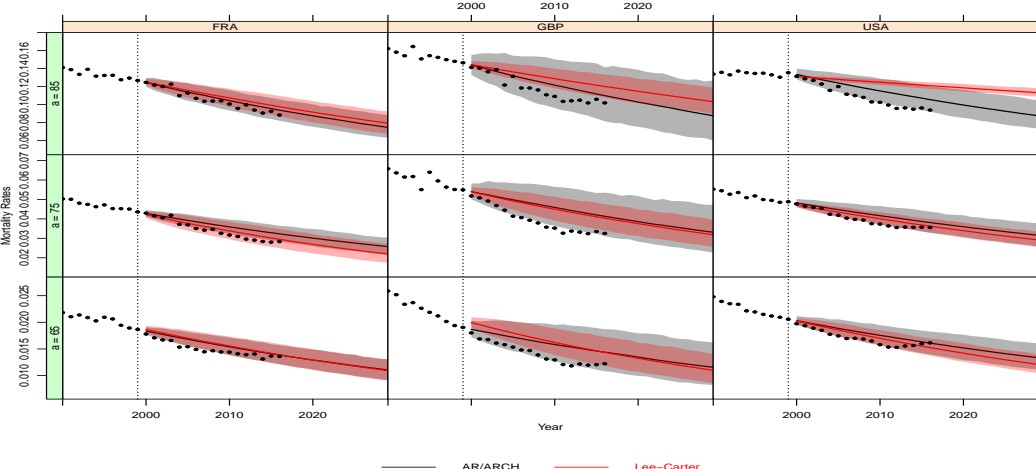

**Figure A5.** Mortality rates: Forecast mortality rates over the period of 2000–2030 at ages 65, 75, and 85 for the France, England and Wales, and US populations produced by both the selected AR-ARCH (black) model and Lee–Carter (red) model. The 95% prediction intervals are also included. The crude mortality over the observed period is also presented (black circles).

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
