# Peer review of "Optimal Neighborhood Selection for AR-ARCH Random Fields with Application to Mortality"

_stats, doi:10.3390/stats5010003_

Round 1

Reviewer 1 Report

In this paper, the authors address the model selection problem when the AR-ARCH process is used to model the mortality rates. In general,  the paper is well written, and the topic is worth to publish. I have two issues that the authors should consider in the new version.

  1. Why did the authors introduce an AR-ARCH model instead of an ARMA-GARCH? It is well-known that the ARMA(p,q)-GARCH(p1,q1) process is more flexible to reproduce the dependence structure observed in the data even if few parameters are considered.
  2.  The recent literature about the mortality models [1,2] shows that the autocorrelation function of the mortality rate is not a monotonic decreasing function. I would like to see which type of autocorrelation can be reproduced by the proposed model. The authors can provide different shape of the autocorrelation using only simulated data. It is not necessary to determine a closed form formula for the autocorrelation function.

Bibliography:

1] Hitaj, A., Mercuri, L. & Rroji, E. Lévy CARMA models for shocks in mortality. Decisions Econ Finan 42, 205–227 (2019). https://doi.org/10.1007/s10203-019-00248-9.

2] Anatoliy Swishchuk, Rudi Zagst, Gabriela Zeller,
Hawkes processes in insurance: Risk model, application to empirical data and optimal investment, Insurance: Mathematics and Economics,
Volume 101, Part A, 2021, Pages 107-124,

Reviewer 2 Report

Please see attached the comments.
